

# A leaf area index, LAI, data set acquired in Sahelian rangelands (Gourma, Mali) over the 2005-2017 period.

Eric Mougin[1], Mamadou Oumar Diawara[2], Nogmana Soumaguel[3], Ali Amadou Maïga[3], Valérie Demarez[4], Pierre Hiernaux[5], Manuela Grippa[1], Véronique Chaffard[6], Abdramane Ba[7]

*Correspondence to*: Eric Mougin (eric.mougin@get.omp.eu)

[1]Géosciences Environnement Toulouse (GET), (UMR 5563, CNRS, IRD, Université Paul Sabatier), Observatoire Midi-Pyrénées, 14 Avenue Edouard Belin, 31400 Toulouse, France. Email : eric.mougin@get.omp.eu

[2]Département de Biologie, Faculté des Sciences et Techniques (FST), Université des Sciences, des Techniques et des Technologies de Bamako (USTTB), Colline de Badalabougou, B.P.3206 Bamako, Mali. Email : diaprod@hotmail.com

[3]Centre IRD, Quartier Hippodrome, BP 2528, Bamako, Mali. Email: nogmana.soumaguel@ird.fr; alimaiga1976c@yahoo.fr

[4] Centre d'Études Spatiales de la Biosphère (UMR 5126, CNES CNRS IRD Université P. Sabatier Toulouse III), 18 avenue Edouard. Belin, 31401 Toulouse cedex 4, France. Email : valerie.demarez@cesbio.cnes.fr

[5]Pastoc, 30 chemin de Jouanal, 82160 Caylus, France. Email: pierre.hiernaux2@orange.fr

[6]Institut des Géosciences de l'Environnement (UMR 5001, CNRS IRD Université Grenoble Alpes (UGA) et Grenoble-INP

38058 Grenoble, France. Email : veronique.chaffard@ird.fr

[7] Laboratoire d'Optique, de Spectroscopie et des Sciences Atmosphériques (LOSSA), Département de Physique. Faculté des Sciences et Techniques (FST), Université des Sciences des Techniques et des Technologies de Bamako (USTTB), BP E3206, Badalabougou, Bamako, Mali. Email : abdramaneba55@yahoo.fr





**Abstract.** The leaf area index, LAI, of Sahelian rangelands and related variables such as the vegetation cover fraction, fCover, the fraction of absorbed photosynthetically active radiation, fAPAR, and the clumping index, $\lambda_o$, were measured in the Gourma region (Mali) during 13 successive rainy seasons, between 2005 and 2017. These variables, known as climate essential variables, were derived from the acquisition and the processing of hemispherical photographs taken along 1 km linear sampling

5    transects, for 5 contrasted canopies and one millet field. The same sampling protocol was applied in a seasonally inundated *Acacia* open forest, along a 0.5 km transect, by taking photographs of the understorey and the tree canopy. These observations collected over more than a decade, in a remote and not very accessible region, provide a relevant and unique data set that can be used for a better understanding of the Sahelian vegetation response to the current rainfall changes. The collected data can also be used for satellite product evaluation and land surface model development and validation.

10    **DOI of the referenced dataset:** doi:10.17178/AMMA-CATCH.CE.Veg_Gh



## 1. Introduction

The Global Climate Observing System, GCOS, identified the leaf area index, LAI, that is the one half the total green leaf area per unit horizontal ground surface area (Chen and Black, 1992; Watson, 1947) as one of the main terrestrial essential climate

variables, ECVs, to be monitored from systematic long-term satellite and in situ measurements (GCOS, 2011). Leaf area index, LAI, or plant area index, PAI, when all the above-ground areal extent of green vegetation are considered, is related both to the fraction of absorbed photosynthetic active radiation (0.4–0.7 μm) by the green vegetation, fAPAR (Myneni and Williams, 1994; Fensholt et al., 2006), and to the canopy cover fraction, fCover, i.e. the amount of vegetation distributed in a horizontal plane (Carlson and Ripley, 1997). These ECV drive the fundamental physiological processes at leaf and canopy levels, such

as photosynthesis and transpiration, as well as the energy and mass exchanges between the surface and the atmosphere (Sellers, 1985). The seasonal variation of the ECVs, i.e. the vegetation phenology, plays an important role in the modulation of energy, water and gas exchanges (Boulain et al., 2009; Jarlan et al., 2008), surface properties such as surface albedo (Guichard et al., 2009) and evaporation-transpiration partitioning (Wang et al., 2014). As such, these ECVs are key variables required in most land surface and biogeochemical models (Potter et al., 1993; Running and Gower, 1991; Wang et al., 2007) or production

efficiency models (Mougin et al., 1995; Running et al., 2004; Tracol et al., 2006; McCallum et al., 2009).

The Sahel is a vast ecoclimatic and biogeographic region extending south of the Sahara on 6000 km long and 400-600 km wide, from the Atlantic Ocean to the Red Sea (Le Houérou, 1989). The climate is arid to semi-arid tropical with a 1 to 4 month single rainy season centred in July – August, and controlled by the West African monsoon, WAM, (Nicholson, Tucker, & Ba, 1998; Lebel et al., 2003). Annual rainfall ranges between approximately 100 mm in the Northern edge of the Sahel to

approximately 600 mm at the Southern edge with the humid Sudanian savannahs. The vegetation comprises a herbaceous layer almost exclusively composed of annual plants, among which grasses dominate, and scattered shrubs and low trees (Hiernaux et al., 2009a). During the short rainy season, the grass layer develops rapidly following the first rains which occur between late May and mid-June, and the herbaceous production is achieved within a few weeks, peaking between late August and mid-September (Breman and Cissé, 1977; Penning de Vries and Djiteye, 1982). Afterwards, at the end of the rainy season in

September, the herbaceous canopy suddenly dries out and, consequently, the amount of green vegetation decreases rapidly with plant wilting (Hiernaux et al., 2009a). Herbage production, and therefore LAI, fCover and fAPAR values, are characterized by high spatial and inter-annual variations that are related to the variability of the WAM. After a humid period in the 1950s, the Sahel experienced a long period of below average rainfall regional conditions (Lebel et al., 2009; Nicholson, Tucker, & Ba, 1998) that was punctuated by major droughts in 1972-73 and 1983-84 with dramatic consequences on the

natural resources, and the population (Stige et al., 2006).

The Sahel is not a very accessible region and in situ measurements of LAI, fAPAR and fCover are scarce, and were mainly performed in millet fields (Hanan and Bégué, 1995; Levy and Jarvis, 1999; Soegaard et al., 1999; Boulain et al., 2009), with a few data acquired in fallows (Boulain et al., 2009) and rangelands (Mougin et al., 2014). Since 1998, moderate spatial resolution (typically 0.5 km to 1 km resolution) sensors such as the Moderate Resolution Imaging Resolution

Spectroradiometer, MODIS, the SPOT-VEGETATION, and the PROBA-V (François et al., 2014) instruments have been providing continuous estimations of the ECV variables at a global scale (Myneni et al., 2002; Baret et al., 2013; Verger et al., 2015). However, the proposed products still need to be evaluated using reliable field estimates (Privette et al., 2002; Sea et al., 2011; Yan et al., 2016a, 2016b; Xu et al., 2018a; 2018b). Particularly, several studies have pointed out the need to sample ecosystems like the semi-arid rangelands characterized by small, spatially heterogeneous and highly variable seasonally LAI

values at the scale of moderate satellite resolution (e.g. Morisette et al., 2006; Baret et al., 2006).

With the main objectives to validate satellite vegetation products, such a rangeland site, located in Mali, in the Sahelian Gourma region, has been integrated in 2000, among the site network used in the Validation of Land European Remote Sensing Instruments, VALERI project (Baret et al., 2005; www.avignon.ina.fr/valeri/; Camacho et al., 2013) and the Committee on



Earth Observation Satellites (CEOS)/Land product validation (Morisette et al., 2006; http://lpvs.gsfc.nasa.gov/) projects. Among the selected validation sites, the Gourma site stands out as the one with the lowest spatial heterogeneity when high (SPOT) and moderate spatial resolution (MODIS) products are compared (Garrigues et al., 2008). Before the set-up of a seasonal and inter-annual survey, two preliminary field campaigns took place in 1999 and 2000, followed since 2005 by a

systematic monitoring of eight 1 km x 1 km sites, during the Sahelian herbaceous growing period matching approximately with the rainy period, from June to September.

## 2. Site description

The vegetation sites are located in north-east Mali, in the Gourma region which stretches from the loop of the Niger River

southward down to the border with Burkina-Faso. Since 2011, due to persistent security problems in the region, the monitored sites were restricted to the vicinity of the Hombori town (15.3°N, 1.5°W) within the 50 km x 50 km supersite (Figure 1) of the African Multidisciplinary Monsoon Analysis (AMMA) - Cycle Atmosphérique et Cycle Hydrologique (CATCH) observing system (Lebel et al., 2009; 2010; Mougin et al., 2009a; Galle et al., 2018; Gaillardet et al., 2018). At the national meteorological station of Hombori, the annual rainfall mean was 373 mm over the 1926-2017 period, and the mean annual temperature was

30.2°C (1950-2010).

The LAI measures were carried out in the sites previously installed in 1984 and monitored till 1994 by the International Livestock Centre for Africa (ILCA) and by the Institut d'Economie Rurale (IER, Bamako) (Hiernaux et al., 1984; Hiernaux et al., 2009a; 2009b), and reactivated by the AMMA-CATCH observing system during the AMMA project (Redelsperger et al., 2006). These 1 km x 1 km sites were chosen within large and relatively homogeneous areas, to sample the main vegetation

types and canopies encountered within the super-site: herbaceous canopies on sandy soils, seasonally inundated *Acacia* open forest and its understory herb layer in a clayed-loamy plain, and a seasonally inundated herbaceous canopy in a clayed loamy plain. In addition, a few data were acquired at an erosion surface site and in a crop millet field, for comparison purposes. Eventually, a total of 8 sites on various soils, were retained, corresponding to 9 vegetation canopies (Tables S1- S2 and Figures S1-S2). The selected vegetation canopies are as follows:

- 4 herbaceous canopies on sandy soils: *Agoufou* (#17, Ag), *Timbadior* (#18, Ti), *Hombori Hondo* (#19, Hh) and *Tara* (#31, Ta)
- 1 herbaceous canopy in a clay loamy plain: *Kelma plain* (#21b, Kp)
- 1 *Acacia seyal* open forest in a clay-loamy plain: *Kelma forest* (#21, Kf)
- 1 understory herbaceous canopy in the *Acacia seyal* forest, *Kelma herbs* (#21, Kh)
- 1 herbaceous canopy on an erosion surface: *Eguerit* (#40, Eg)
- 1 millet field on sandy soil: *Bilantao* (#41, Bi)

## 3. Field sampling strategy and data description

The temporal variation of in situ LAI, $\lambda_o$, fCover and fAPAR were estimated through the analysis of digital hemispherical photographs (Weiss et al., 2004; Mougin et al., 2014). These photographs were collected during the growing season of

herbaceous canopies or over the whole year for the forest canopy (Figure 2a, 2c), and regularly taken along one or two perpendicular 1 km sampling transects crossing the site, usually in the North-South or the East-West directions (Table S3). At the seasonally inundated forest site, the unique field transect was limited to 0.5 km due to the difficulties associated with the field work in such an environment. At each site, the starting and ending points of the transects were geolocated using global positioning systems (Garmin Trex©) to within an approximately 10-metre accuracy. At each sampling date, approximately 100

(50) hemispherical photographs were acquired at the herbaceous (forest) sites, that is a picture taken every approximately 10 meters. At the forest site, photographs were acquired both in the upward and downward directions to sample the forest canopy





and the herbaceous understorey. When the forest floor was inundated, which could occur during the rainy season, only the herbaceous vegetation component above the water surface was considered.

All the pictures were taken using a Nikon Coolpix 8400 (8 Mp) equipped with the fisheye FC-E9, viewing vertically downward above the herbaceous vegetation canopy with a distance between the top of the canopy and the lens of approximately 40 cm,

providing a sampling area of approximately 1 m² (Mougin et al., 2014). At the *Acacia* forest site, the camera was mounted on a levelled tripod, and orientated upward at a height of approximately 0.8 m above the forest floor and maintained horizontal thanks to bubble-levels.

Prior to the field campaigns, the camera was calibrated using the method described in http://www.avignon.inra.fr/can_eye, to compute the optical centre of the camera–fisheye lens system (Demarez et al., 2008). The collected hemispherical pictures

were analysed using the image processing software CAN-EYE V5.0 which is dedicated to the classification of green and non green vegetation elements, and to series of photographs (Baret and Weiss, 2004). The CAN-EYE software provides LAI estimates from gap fraction measurements derived from the hemispherical photographs (Figure 2b, 2d), and has proved to be efficient to derive the vegetation variables of herbaceous (Demarez et al., 2008; Mougin et al., 2014) and forest (Sea et al., 2011) canopies.

The main outputs of the CAN-EYE software are the leaf area index, LAI, the aggregation factor or clumping index $\lambda_o$, the vegetation cover, fCover, and the daily fraction of integrated absorbed photosynthetically active radiation, fAPAR (Table 1).

Finally, the estimated mean vegetation variables at the 1 km scale, are computed by averaging all the 100 or 50 measurements acquired along the sampling transect for the herbaceous and forest canopy, respectively. The performance of the combined use

of hemispherical photographs, and the CAN-EYE software for estimating LAI, $\lambda_o$, fAPAR and fCover at the 1 km transect scale has been analysed by Mougin et al. (2014) for a herbaceous canopy. These variables are estimated at a spatial scale which is compatible with remote sensing observations performed by moderate resolution sensors. Particularly, the clumping index brings valuable information on the temporal variation of the heterogeneity of grass cover, and can be retrieved from remote sensing observations (Pisek et al., 2013; He et al., 2016). The estimated accuracy associated to the field estimation of LAI is

approximately ±17.3% for the herbaceous canopies (Mougin et al., 2014), and ±36.5% for the PAI estimates at the *Acacia* forest, when the errors of classification and spatial sampling are considered (this study).

Whenever possible, hemispherical photographs were taken approximately every 10 days during the growing seasons for the herbaceous canopies whereas at the *Kelma* forest site, the monitoring took place approximately every 10 days during the leafy period i.e. from July to January, and otherwise every month during the dry season. An illustration of the high seasonal temporal

variation of LAI for 5 monitored herbaceous canopies, is given in Figure 3 for the 2017 growing season. The strong temporal dynamics of the herbaceous cover justifies the 10-days resolution sampling to catch the seasonal vegetation cycle, particularly the maximum LAI, $LAI_{max}$, which is usually observed towards the end of the rainy season. Afterwards, the herbaceous green canopy rapidly dried out apart from the forest understorey composed partly with the perennial herb *Sporobolus helvolus*, in the loamy-clayed soil (#21, *Kelma* herbs), which benefited from more humid conditions. Large differences in $LAI_{max}$ values

are observed among sites depending on soil fertility, soil water budget, intensity of grazing by livestock, and length of soil inundation in the clayed sites in depressions (# 21 and # 21b). In 2017, maximum LAI values, $LAI_{max}$, ranged between 0.43 at the seasonally flooded *Kelma* site, and 1.58 at the *Agoufou* sandy site.

The magnitude of the inter-annual variability in LAI for herbaceous canopies, observed over the 13-year monitoring period is quite high (see the example given in Figure 4a), with temporal coefficients of variation higher than 40% for the herbaceous

sites, up to approximately 100% for the forest understorey (Table 2). Over the 2005-2017 period, the mean (s.d.) LAI value for the herbaceous sites on sandy soils was 0.96 (0.27).

Similarly, the observed PAI of the *Kelma* forest canopy exhibits large inter-annual variations, with maximum PAI values varying from 0.95 in 2005 to 2.6 in 2014, corresponding to a foliage area index, LAI, of approximately 0.64 to 2.29 (Figure



4b) once the wood area index, WAI, estimated at 0.31 by averaging PAI values during the leafless period, has been subtracted from the PAI estimates. Over the 2005-2017 period, the coefficient of variation of maximum PAI values of the *Kelma* forest is approximately 35.4 %, a value close to the 41.3% estimated along the *Agoufou* EW transect for the herbaceous $LAI_{max}$.

## 4.    Data base and data availability

The dataset presented in this paper, is part of the AMMA-CATCH observatory database (www.amma-catch.org), and is identified by the permanent identifier doi:10.17178/AMMA-CATCH.CE.Veg_Gh, which describes its content with standardized metadata (DataCite[1]), and gives access instructions. Table 3 shows the data availability included in this dataset: it covers the 2005-2017 period and is given for the 13 monitored transects. The dates of hemispherical photograph acquisitions are given in Table S4. Over the 13-year period, a total of 658 sampling transects have been monitored, and approximately 52 000 hemispherical images have been acquired, processed and analysed. Associated data such as herbaceous aboveground masses, $CO_2$, sensible and latent flux measurements, meteorological observations and soil moisture measurements collected at the *Agoufou*, *Kelma* and *Eguerit* sites can also be found in the AMMA-CATCH database.

The access to the data is available online, free of charge on the web AMMA-CATCH database portal: http://bd.amma-catch.org and without any barriers except a registration to get a login free-of-charge. Sign up is only required to download the data but not to browse the content of the datasets (metadata include variables names, spatial extent and coordinates, temporal extent, contacts, ...), or to retrieve their content in a ISO 19115[2] standard file, or to localise on the map the acquisition sites. To access to data download, once connected to the portal, search for the dataset by identifying it by its doi suffix (CE.Veg_Gh) or title among all the datasets contained in the database, and unfold the tab describing it. The database architecture also allows to extract a subset of the data at the spatio-temporal or variable level. The data files are provided in two formats: ASCII csv and NetCDF.

The data is licensed under the Creative Common Attribution 4.0 International Licence[3] (CC-BY 4.0) that sets the conditions of the data use. For any publication using AMMA-CATCH data, depending on the contribution of the data to the scientific results obtained, data users should either propose co-authorship to the dataset Principal Investigators or at least acknowledge their contribution.

## 5.    Use of the data set

The main objective of the monitoring is to document the seasonal, inter-annual and decadal variations of LAI and thus primary productivity in relation to the rainfall variability, at different spatio-temporal scales. More precisely, the current monitoring aims to provide relevant data to investigate how the WAM and its spatio-temporal variability affect the vegetation cover in northern Sahel. This data set can also be used in the interpretation of Sahelian surface processes that are mainly modulated by the vegetation cover such as the partition between latent and sensible energy fluxes (Timouk et al., 2009), dust and biogenic chemical emissions from soils (Pierre et al., 2012; Delon et al., 2015). In addition, examples of data use in validation exercises of remote sensing products and surface models, are briefly presented below.

---

[1] DataCite: Locate, identify, and cite research data, https://www.datacite.org/

[2] ISO 19115: Geographic information – Metadata standards, https://www.iso.org/standard/53798.html

[3] The following sentence should appear in the acknowledgments of the publication: "The AMMA-CATCH regional observing system (www.amma-catch.org) was set up thanks to an incentive funding of the French Ministry of Research that allowed pooling together various pre-existing small scale observing setups. The continuity and long-term perennity of the measurements are made possible by an undisrupted IRD funding since 1990 and by a continuous CNRS-INSU funding since 2005".





### 4.1. Validation of satellite products

Validation of existing LAI products is still a challenge (Xu et al., 2018a; 2018b) and a homogeneous long-term LAI data set, based on an unchanged field protocol during more than a decade, is certainly of valuable interest with the objectives of validating satellite products (Fang et al., 2012). The spatial sampling along transects of 1 km, as performed during the

present monitoring, has also been highly recommended for field-based validation of moderate resolution products (Baret et al., 2005; Sea et al., 2011). The temporal resolution of the sampling, i.e. 10 days, should allow LAI products to be accurately evaluated. Moreover, the corrections of the clumping effects performed thanks to the estimated aggregation factor, enable another source of uncertainty to be considered, linked to the non-random green components distribution within the canopy (Chen et al., 1997; Kucharik et al., 1998).

Preliminary LAI and fCover 1-km estimates at the *Timbadior* site, with a different technique, have already been used in direct validation exercises using the VEGETATION products within the frame of the VALERI project (Baret et al., 2005; Weiss et al., 2007; Fang et al., 2012), the MODIS products (Mougin et al., 2009b), and the Global LAnd Surface Satellite, GLASS, fCover products (Xiao et al., 2016). In this paper, we present two illustrations of a similar exercise (Diawara et al., in preparation), using the LAI/fAPAR field data acquired during the 2005-2017 period, at the *Agoufou* savannah (Figure 5a)

and *Kelma* forest (Figure 5b) sites, and the latest MODIS products, (MOD15A2H, Collection 6), available since 2015 at a spatial resolution of 500 m (Myneni and Park, 2015). The Collection 6 benefited from improved surface reflectances and biome type inputs (Wolfe et al., 2013), and provided more accurate products (Xu et al., 2018b).

### 4.2. Validation of land surface models

The LAI dataset was also used to evaluate simulations by dynamic vegetation models, particularly those used in the AMMA Land Surface Models Intercomparison Exercise (ALMIP, Boone et al., 2009a, 2009b; Grippa et al., 2017). This project was carried out to evaluate the capability of different state of the art land surface models to reproduce surface processes in West Africa. The second phase of ALMIP focused on the three meso scale sites in Benin, Niger and Mali, instrumented by the AMMA-CATCH observatory (Galle et al., 2018), that provides specific data for forcing and evaluating model outputs. An

example of the comparison between in-situ LAI and simulated LAI by the Sahelian-Transpiration-Evaporation-Productivity model, STEP, (Mougin et al., 1995, Pierre et al., 2016) for the ALMIP-2 runs is reported in Figure 6. Both the seasonal and the interannual variability in LAI, including water stress events such as that happening in 2008, are well reproduced by STEP. However, the comparison with field data shows that the model response is a bit smoother than in the in-situ dataset.


### 5. Conclusion

A 13-year consistent leaf area index (LAI) data base was built for the 2005-2017 period, in a not easily accessible and weakly sampled semi-arid region, in Northern Sahel. The data set includes additional essential climatic variables such as the vegetation cover fraction, the fraction of absorbed photosynthetically active radiation, and the clumping index. All these variables were

estimated from digital hemispherical photographs taken along 0.5 or 1 km transects crossing different sites in the Gourma region in north-east Mali. The same processing software, CAN-EYE, was used for image classification, gap fraction analysis and LAI derivation providing a homogeneous and consistent data set. Among others, this data set can be used to validate moderate resolution satellite products, to establish relationships between satellite vegetation indices and vegetation variables, to evaluate land surface models, to interpret surface fluxes and above all to study the decadal response of Sahelian ecosystems

to the rainfall variability of the West African monsoon.



**Supplementary material**

Supplementary data sets related to AMMA-CATCH instruments, including meteorological, flux, soil moisture, herbaceous aboveground mass, tree species composition and woody plant population densities data can be downloaded from the AMMA-CATCH data base (http://www.amma-catch.org/spip.php?rubrique63).

**Author contribution**

Data sets were collected by E. Mougin, M.O. Diawara, N. Soumaguel, A. Maïga, V. Demarez and P. Hiernaux. M.O. Diawara and E. Mougin prepared the data sets with the help of V. Chaffard who oversees the AMMA-CATCH data base. All co-authors contributed to the criticizing of the individual datasets and contributed to the writing of the

10   manuscript.

**Acknowledgements**

Many thanks are expressed to the numerous people who participated in the field data acquisition: Ali Maïga and Tarawetti Ag Mostafa from Hombori, Hamma Maïga from Agoufou, Youssouf Maïga† from Gao, Alexis Berg,

15   Marie-Noëlle Mulhaupt, Téri Massa-Mabilo and Yann Tracol. We also thank Anne-Marie Cousin for her help for the figures.

†deceased





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





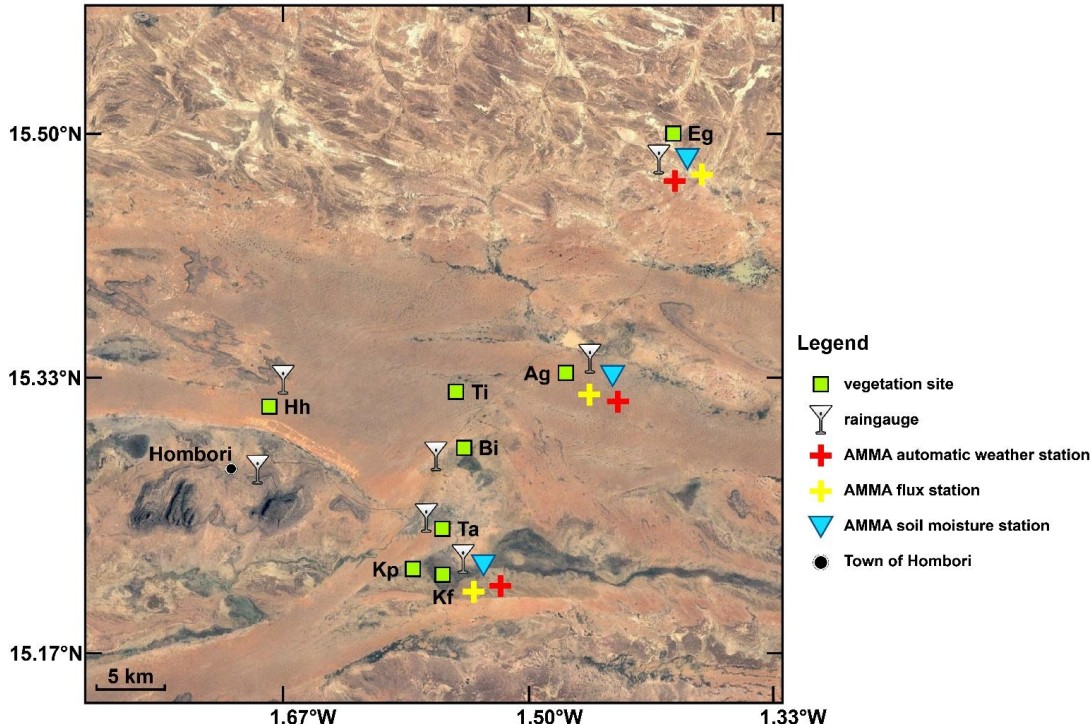

**Figure 1.** Map (on a Google-Earth image, 2016/31/12) of the Hombori supersite showing the location of the 8 vegetation canopies: *Agoufou* (#17, Ag), *Timbadior* (#18, Ti), *Hombori Hondo* (#19, Hh), *Tara* (#31, Ta), *Kelma forest* (#21, Kf) and *Kelma plain* (#21b, Kp), *Eguerit* (#40, Eg), *Bilantao* (#41, Bi), the national meteorological station at Hombori and the automatic meteorological, flux and soil moisture stations installed during the AMMA project during the 2005-2010 period. The site code number refers to Hiernaux et al. (2009a).

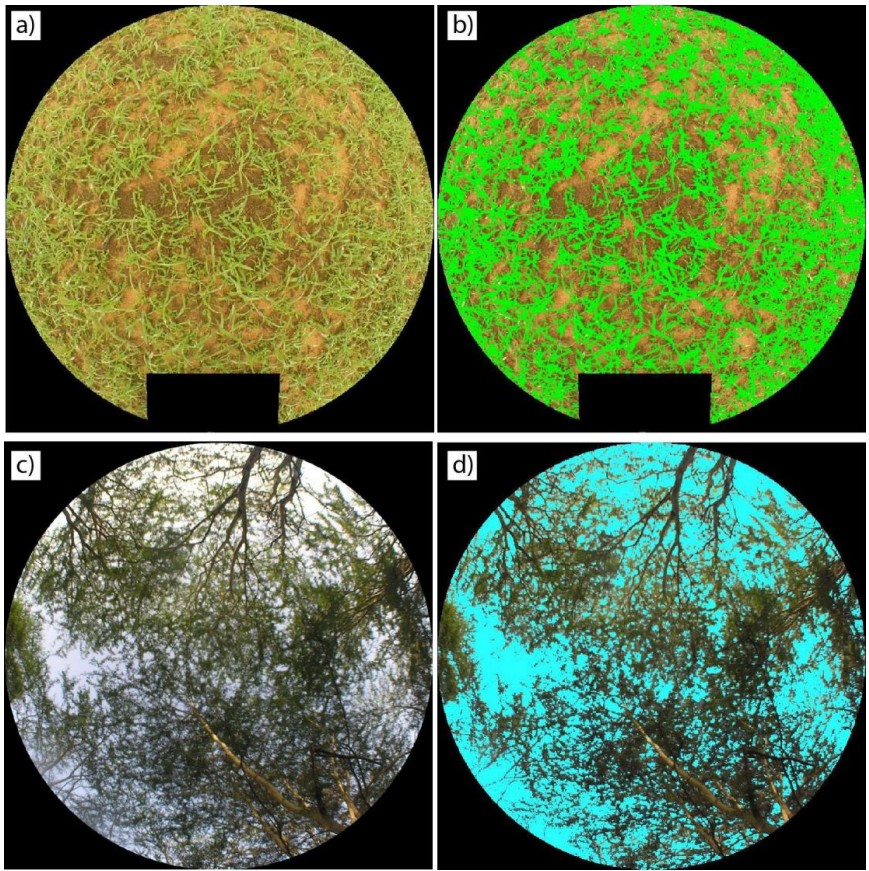

**Figure 2. Hemispherical photographs taken approximately at the maximum of green vegetation during the rainy season, and the corresponding classified images: (a) and (b) herbaceous canopy (*Agoufou*, 2007/07/29); (c) and (d) forest canopy (*Kelma fores*t, 2016/09/29). Black zones on the (a) and (b) images, correspond to masked areas. The processing of hemispherical images is restricted to 0–60°○zenith angles to avoid mixed pixels. In the case of the forest canopy, note that the process could classify either vegetation elements like green herbaceous leaves (b), or non vegetation elements like sky pixels (d).**





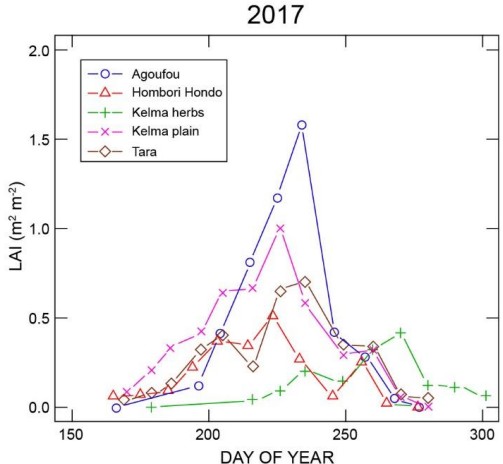

**Figure 3. Seasonal variations of the leaf area index (LAI) of herbaceous canopies, observed in different sites from May to**
5   **December 2017.**




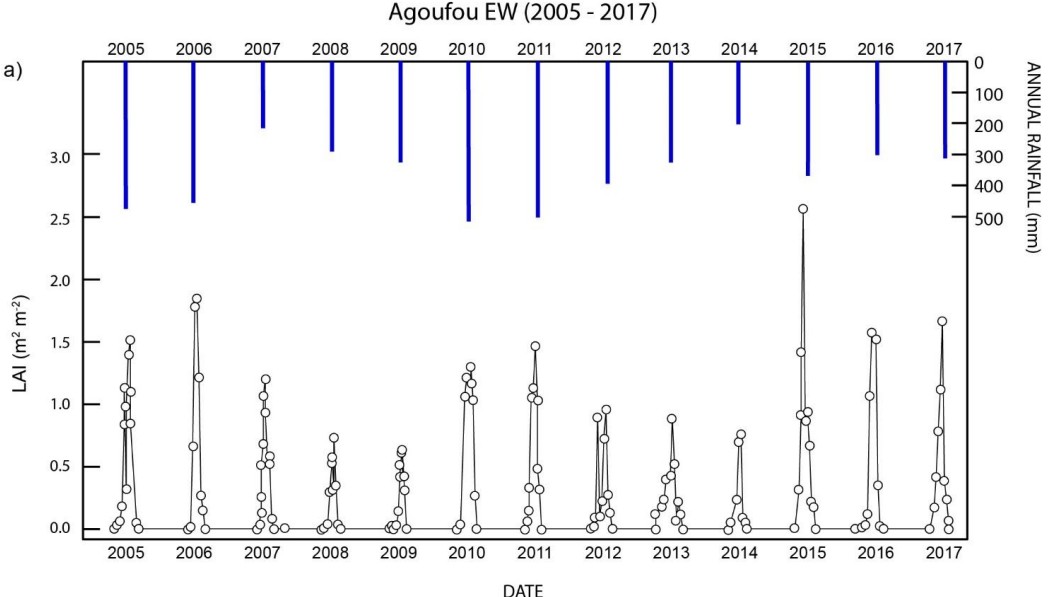

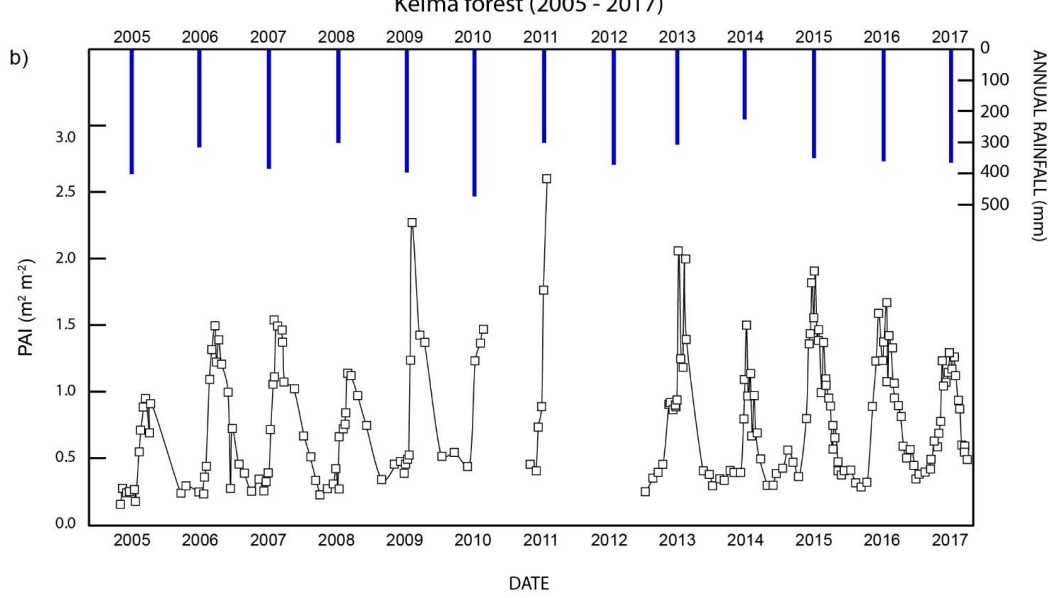

**Figure 4. Inter-annual variations of a) the herbaceous leaf area index (LAI) at the *Agoufou* site, and b) the plant area index (PAI) of the *Kelma* forest canopy recorded during the 2005 – 2017 period. Note that no data have been acquired from September 29, 2010 to June 06, 2011 and from September 08, 2011 to March 04, 2013 at the *Kelma forest*, due to security issues. Inter total annual precipitations are also shown.**





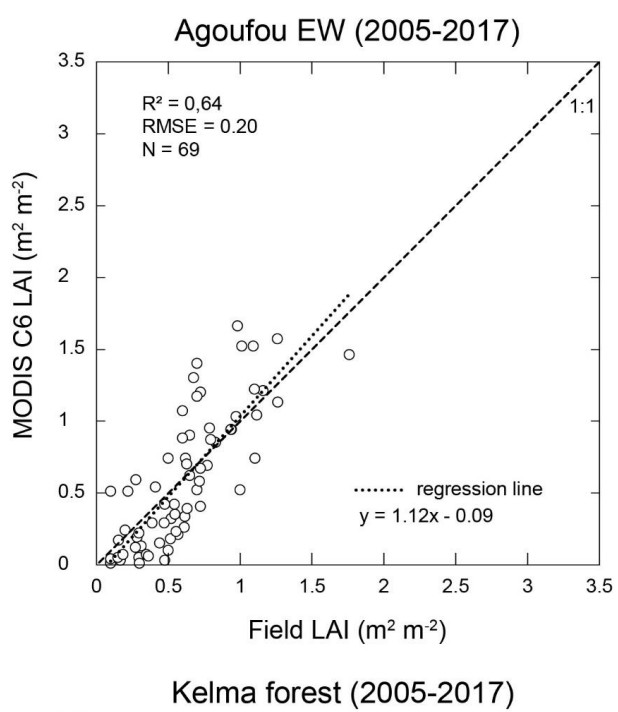

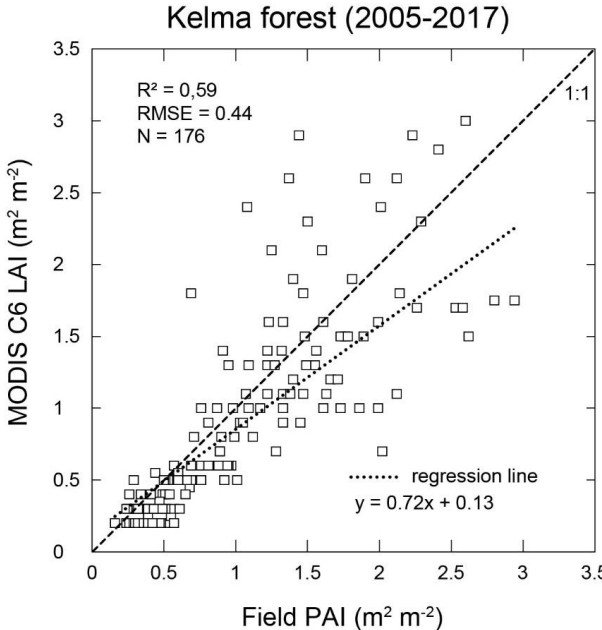

**Figure 5. Comparison between field LAI or PAI estimates and MODIS Collection 6 LAI products for the period 2005-2017: a)**
***Agoufou* herbaceous canopy and b)** *Kelma forest* **site. In this later case, the MODIS LAI product is compared to the sum of the tree**
5    **PAI, and the LAI of the understorey.**




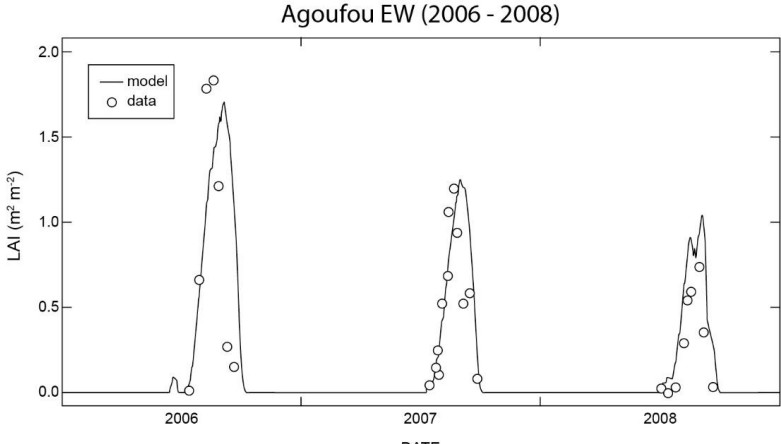

5     **Figure 6. Comparison between field LAI measurements (+) at the *Agoufou* site and simulated LAI (-) using the STEP model (2006-2008) during the ALMIP-2 exercise (Boone et al., 2009a; 2009b).**





| Variable | Symbol | Unit |
|---|---|---|
| **Leaf Area Index** | LAI | $m^2\ m^{-2}$ |
| **Clumping Index** | $\lambda_o$ | no unit |
| **Vegetation Cover Fraction** | fCover | $m^2\ m^{-2}$ |
| **Daily fraction of absorbed photosynthetic active radiation** | fAPAR | $MJ\ MJ^{-1}$ |

**Table 1: List of the vegetation variables estimated along the 1 km or 0.5 km transects.**





| Site # | Start (year) | Years (n) | Mean (s.d.*) LAI$_{max}$ (m$^2$ m$^{-2}$) | Maximum LAI$_{max}$ (m$^2$ m$^{-2}$) | Minimum LAI$_{max}$ (m$^2$ m$^{-2}$) | Coefficient of variation (%) |
|---|---|---|---|---|---|---|
| Agoufou (#17) NS | 2007 | 5 | 1.18 (0.82) | 2.36 | 0.57 | 69.5 |
| Agoufou (#17) EW | 2005 | 13 | 1.31 (0.54) | 2.56 | 0.63 | 41.3 |
| Timbadior (#18) NS | 2007 | 6 | 1.19 (0.66) | 2.27 | 0.57 | 55.6 |
| Timbadior (#18) EW | 2007 | 3 | 0.91 (0.53) | 1,52 | 0,58 | 58.1 |
| Hombori Hondo (#19) NS | 2007 | 11 | 0.7 (0,27) | 1.25 | 0.34 | 38.0 |
| Hombori Hondo (#19) EW | 2007 | 3 | 0,55 (0.06) | 0.60 | 0.48 | 11.6 |
| Kelma herbs (#21) EW | 2005 | 13 | 1.01 (0.98) | 3.70 | 0.17 | 97.4 |
| Kelma plain (#21b) EW | 2008 | 9 | 1.06 (0.57) | 1.91 | 0.10 | 53.5 |
| Tara (#31) NE-SW | 2007 | 11 | 1.10 (0.71) | 2.52 | 0.33 | 65.0 |
| Tara (#31) NW-SE | 2008 | 2 | 0.58 (0.19) | 0.71 | 0.44 | 33.2 |
| Eguerit (#40) E-W | 2008 | 1 | 0.08 | - | - | - |
| Bilantao (#41) NE-SW | 2007 | 4 | 0.82 | 1.24 | 0.37 | 44.6 |

*standard deviation

Table 2: Statistics of the maximum seasonal LAI, LAI$_{max}$, of the herbaceous canopies over the 2005-2017 period. The number of
5   monitored growing seasons (years) per site is also indicated.



| Site # | 2005 | 2006 | 2007 | 2008 | 2009 | 2010 | 2011 | 2012 | 2013 | 2014 | 2015 | 2016 | 2017 | Total |
|---|---|---|---|---|---|---|---|---|---|---|---|---|---|---|
| Agoufou NS (#17) | | | 3 | 5 | 8 | 1 | 6 | | | | | | | 23 |
| Agoufou EW (#17) | 13 | 8 | 12 | 10 | 10 | 7 | 9 | 9 | 10 | 6 | 9 | 8 | 8 | 119 |
| Timbadior NS (#18) | | | 5 | 5 | 5 | 2 | 4 | | | 1 | | | | 22 |
| Timbadior EW (#18) | | | 2 | 4 | 4 | | | | | | | | | 10 |
| Hombori Hondo NS (#19) | | | 4 | 4 | 5 | 2 | 2 | 3 | 8 | 7 | 8 | 9 | 11 | 63 |
| Hombori Hondo EW (#19) | | | 2 | 3 | 4 | | | | | | | | | 9 |
| Kelma forest EW (#21) | 12 | 15 | 16 | 17 | 10 | 6 | 6 | | 14 | 18 | 22 | 22 | 29 | 187 |
| Kelma herbs EW (#21) | 6 | 8 | 7 | 5 | 4 | 2 | 3 | 11 | 5 | 6 | 11 | 10 | 9 | 87 |
| Kelma plain EW (#21b) | | | | 1 | 5 | | 3 | 11 | 11 | 6 | 9 | 6 | 11 | 63 |
| Tara NE-SW (#31) | | | 2 | 6 | 6 | 2 | 3 | 3 | 9 | 7 | 9 | 7 | 11 | 65 |
| Tara NW-SE (#31) | | | | 3 | 1 | | | | | | | | | 4 |
| Eguerit EW (#40) | | | | 1 | | | | | | | | | | 1 |
| Bilantao NE-SW (#41) | | | 1 | 2 | 1 | | | | 1 | | | | | 5 |

Table 3: Data availability (green colour) for the different sites and transects sampled since the start of the monitoring in 2005. The number of transects performed per year, and the total of transects sampled during the entire monitoring period are also indicated per site. The total of transects sampled during the 13 year monitoring period is 658.