# Peer review of "A leaf area index data set acquired in Sahelian rangelands of Gourma in Mali over the 2005-2017 period."

_Earth System Science Data, 2018_

## Referee Comment (RC1) · Anonymous Referee #1 · 22 Mar 2019

**Earth System Science Data Review:**
A leaf area index, LAI, data set acquired in Sahelian rangelands (Gourma, Mali) over the 2005-2017 period.
Eric Mougin, Mamadou Oumar Diawara, Nogmana Soumaguel, Ali Amadou Maïga Valérie Demarez, Pierre Hiernaux, Manuela Grippa, Véronique Chaffard, Abdramane Ba

**Paper Summary**
LAI, FCover, PAR and clumping index of rangelands in the Sahel are presented on a time series of 13 succesive rainy seasons form 1005/2017. The variables have been acquired from hemispherical photographs along 1 km sampling. The paper also present retrievals of LAI based on moderate radiometers. Data seems to be useful for satellite direct retrievals of LAI and other variables and or for model assessment and comparisons.

**Review Assessment**
This paper contains a number of errors that makes the reading flow difficult. Now it also contains numerous spelling and technical errors. However, the most important question I have is what is the science here? I understand the value of some of the variables but what is the science here is hard to get. This review finds this paper border line between rejection and major reviews. This is because this journal seems to focus on data. However, science data should be explained in the more compelling and more justifiable scientific framework.

**Comments**

Delete LAI from the title.
Change rangelands of Gourma in Mali – Eliminate parenthesis.

Through the abstract there are three instances in which you define acronyms that are not used through the abstract. - then I would suggest to use the entire word and then when the paper start in the Introduction you define those acronyms.

Why these variables are climate essential? You are giving swepping statement without justification. Please restrict you writing to the specific of what the variable is about without loading any extra concept.

Change "understorey" by  understory

Abstract evaluation: this abstract si unspecific. The abstract should summarize overarching objective, what the project is about and what the main results are. Your abstract does not respond to any of those parameters.

Line 3: explain what you mean by "one half the total leaf area index".

Line 4-off: References should go in chronologic order.

Line 15: what are production efficiency models?

In general acronyms are expressed in parenthesis. Western African Monsoon (WAM).

Line 43: check if you can use an internet link as reference.

Normally at the end of an introduction the paper should describe a roadmap of what the paper is about. Missing here.

Lines 11 and 20: there is a mismatch in the use of supersite and then super-site. You should keep coherency across the paper.

Lines 25 – off change numbers by the actual word.

Line 39: change "metre" by meter

Line 5: give the absolute height to verify the resolution and your camera characteristics.

Line 9: change "centre" by center.

Line 22: moderate remote sensing sensors. You need to put examples and references here.

Line 25: here you express variations of estimates of LAI and PAI. From where these number so are coming from?

Line 31: please use a more academic verb instead of "catch".

Section 4: what is this? I really don't understand the purpose of this section.

Figure 1. This illustrate very-well the sites but it is difficult to see where this is located. Can you please add a larger area map with an inset so that we can see where this region is actually located.

Figure 3. time series of LAI. These series are not clearly explained why Kelma herbs peaks far away from Agoufou?

Figure 4. Here you show rainfall and LAI. However LAI is known to be very sensitive to temperature and therefore inversely dependent to rainfall. Then for datasets that focus on vegetation that eventually respond to climate shift it it mandatory to know the time series of rainfall rather than the accumulated rainfall. The vegetation onset depends not only on radiation and temperature but also on the precipitation during the beginning of the season.

Figure 5. Kelma Forest exhibits a different dynamic behavior than Agoufou for LAI below 1 m2m-2." No explaonation in the text what is the attribution to this feature.

Also the scattering dispersion is larger in the second plot compared to the first pplot for LAI larger than 1 m2m-2.

Figure 6. It is difficult to understand the meaning of this comparison. I think you should clearly explain why this comparison is needed. And. Actually why the series diverge differently.

---

## Referee Comment (RC2) · Anonymous Referee #2 · 25 Mar 2019

First, I would like to appreciate the efforts that have been taken by the authors to collect and analyze 13-years field measurements for the manuscript "A leaf area index, LAI, data set acquired in Sahelian rangelands (Gourma, Mali) over the 2005-2017 period". Secondly, the open-access dataset provided by the authors will be an asset for future research use, which is quite appreciable indeed. However, I believe the manuscript requires major revisions mostly about the way it is written and the structure and text body of especially abstract, introduction, site description, and field sampling strategy and data description. It gets sometimes very difficult to follow the logic and methodology used in this manuscript. I also recommend the authors to fix the sentence structure, e.g., avoid run-on sentences, and grammatical and spelling issues that I have noticed throughout the paper. In addition, I would recommend the authors to rerun the data analysis using yearly MODIS imagery corresponding the year of collected data in the field, which I will explain it later in my list of comments. Furthermore, the current type of provided line numbers makes addressing comments very difficult.

Here are some of my concerns and comments regarding this manuscript:

1.      The title doesn't carry out the main picture and purpose of this research. Also, as I found the study region is in the northern Sahel and thus the title needs to be changed as "…rangelands in the northern Sahel…".

2.      Abstract, P 2, L5-10, you can't draw a strong conclusion about using this dataset for a better understanding of the Sahelian vegetation response to the current rainfall changes unless you scientifically provide evidence that this study area at the northern Sahel has similar vegetation community with the rest of the Sahel.

3.      1. Introduction, L5, please add the exact internet link for this reference (GCOS, 2011) in the citation list and look at the following reference and include it, Bojinski et al. 2014. The Concept of essential climate variables in support of climate research, applications, and policy. Bull. Am. Meteorol. Soc. 95:1431-1443.

4.      P 3 L10, "The seasonal variation of the ECVs, i.e. the vegetation phenology" please change it to "…., i.e. changes in the vegetation phenology".

5.      P3 L20, "low trees" is vague. Does it mean short stature trees or dispersed trees?

6.      P3 L35, MODIS needs to be in parentheses. Please add all abbreviations in the

parenthesis and be consistent about using your terminology.

7.      P3 L35, "…instruments have been providing continuous estimation of the ECV...". Please change it to "… have been used to provide a continuous estimation….".

8.      P3 L35-40, You mentioned past research emphasized on using spatially heterogenous rangeland for validating ECVs on for example MODIS imagery, but at the beginning of next page, you are describing your study site as having lowest spatial heterogeneity. Looks these statements are contradictory. Please explain why your selected study site was different from what literature pointed out about the spatial heterogeneity.

9.      P3 L40, the whole sentence from "With the main objectives….gov/) projects." is a vague run-on sentence. Please break it down to smaller clear sentences. Also, is it the objective of this study? If so, you need to clearly state this objective. I assume you have two main objectives: 1-validation of computed vegetation indices from the satellite imagery such as MODIS 2-providing open-access dataset. You need to deliver these main objectives at the end of introduction in a clear and strong way.

10.     P4 before L5, "Before the set-up of a seasonal …" this sentence is about field sampling strategy and needs to go to that section.

11.     3. Site description, P4 L10, please change "the annual rainfall mean" to "the mean annual rainfall".

12.     P4 L20, what is "super-site"? Please explain it at the first time you are using it.

13.     P4 L20, "clayed loamy" should be "clay loam".

14.     P4 L20, "…and its understory herb layer in a clayed-loamy plain, …". Please change it to "...and its understory herb layer in a plain of dominantly clay loam soils,…".

15.     P4 L25, "clay loamy" and on the next sentence "clay-loamy". Please be consistent about using your terminology and change them to "clay loam".

16.     P.4, L35, why were you bias about your sampling directions at N-S or E-W, and not to randomly choose the transect direction? Please justify it.

17.     P.4, L35, "geolocated" please change it to "georeferenced".

18.     P.4, L35, "to within an approximately 10-metre accuracy" please change it to "with approximately 10-meter horizontal positional accuracy".

19.     P4, L40, "approximately 100 (50)"? 100 (50) is vague. Please explain it.

20.     P4, L40, "herbaceous (forest) sites" is confusing. Please explain it. The term forest in the

parenthesis gives readers the impression of equality to the herbaceous.

21.     P5 L5, "…maintained horizontal thanks to bubble-levels." is vague. What does "thanks" mean here? Please rewrite it in a clear way.

22.     P5 L10, "non green" please change it to "non-green".

23.     P5 L10, "and has proved to be efficient" please change it to "and has been proved to be efficient".

24.     P5 L25, "±17.3%" and "±36.5%" are accuracy or standard deviations? If they are standard deviations, please be careful about using your terminology.

25.     P5 L30, "Afterwards, the herbaceous green canopy rapidly dried out apart from the forest understory composed partly with the perennial herb …." It is confusing. Please rewrite it in a clear way.

26.     5. Use of the data set, P6 L30, this is an objective of this study and needs to go to the end of introduction.

27.     4.1. Validation of satellite products, P7, L5, "…the clumping efforts performed thanks to the estimated aggregation factor, …" what is the meaning of "thanks" here. It is a confusing sentence. Please rewrite it.

28.     P7 L10, why is "VEGETATION" capitalized? Does it convey a specific meaning here? Please explain it.

29.     P7 L10-15, what does it mean "The collection 6 benefited from improved surface reflectances and biome type inputs (Wolfe et al., 2013), and provided more accurate products (Xu et al., 2018b)"? It is confusing. Why didn't you compare year by year computed vegetation indices from MODIS imagery and field dataset for 13 years? How do you deal with the yearly fluctuations of precipitation in such an arid climate that significantly impacts the leaf area index and other vegetation characteristics? Based on your analysis, you are assuming the vegetation indices derived from MOIDS imagery at the period of 2005-2017 are similar to the period of 2015-2017. I recommend you rerun the analysis using year by year comparison between computed vegetation indices from the MODIS imagery and the field survey.

30.     P6 L20, please change "meso scale" to "mesoscale".

31.     P14, Figure 1, Please make a joint map with this figure that shows the study area in the African continent.

32.     P18, Figure 5, the issue of specified comparison of MODIS imagery with field surveys,

for vegetation indices, is pretty much clear on the data points where data are clumped row by row, especially in the regression plot for Kelma forest, and this type of comparison is not statistically true and can cause erroneous results.

33.     P22, Table 3, Why are almost half of your sampling sites include a very low yearly sample size, e.g., sites no. #17 Agoufou NS, #18 Timbadior NS, #18 Timbadior EW, #31 Tara NW-SE, #40 Eguerit EW, and #41 Bilantao NE-SW, and how do you justify drawing conclusion from the results of such an small sample size? What is the reason for not having enough samples at those sites? By having such a small sample size, making conclusion for those sites with specific plant communities and thus a broader conclusion for the entire vegetation communities across the northern Sahel could be questionable.

34.     Supplements, Figure S2, the picture numbering (the letters) doesn't correspond to the letters written in the figure caption.

35.     The type of soils in Table S1 and Table S2 do not look correspond to each other for some sites. For example, Kelma Plain (#21b) site shows a dominant soil type of clay loam in the Table S1 but the percentage of soil particles in Table S2 shows a very low amount of clay particles about 1-2 %. Please check them out and use the common soil textural triangle to find out the soil texture class for every site.

---

## Author Comment (AC1) · 12 Apr 2019

Reviewer #1:

Review Assessment This paper contains a number of errors that makes the reading flow difficult. Now it also contains numerous spelling and technical errors. However, the most important question I have is what is the science here? I understand the value of some of the variables but what is the science here is hard to get. This review finds this paper border line between rejection and major reviews. This is because this journal seems to focus on data. However, science data should be explained in the more compelling and more justifiable scientific framework.

[Figure]

Author response: Yes, you are right. The journal focus on data and the main objective of a 'data paper' like the one we submitted, is to present the data, their potential scientific value, and the associated open data base. We think we fulfilled all these criteria.

Comments

Delete LAI from the title.:

OK, corrected.

Change rangelands of Gourma in Mali – Eliminate parenthesis:

OK, corrected.

Through the abstract there are three instances in which you define acronyms that are not used through the abstract. - then I would suggest to use the entire word and then when the paper start in the Introduction you define those acronyms.

OK, corrected. The acronyms are now defined in the Introduction section.

Why these variables are climate essential? You are giving swepping statement without justification. Please restrict you writing to the specific of what the variable is about without loading any extra concept.

These vegetation variables are among the 13 so-called terrestrial essential climate variables (ECVs) identified by the Global Climate Observing System (GCOS) to be monitored from systematic long term measurements collected by satellite and in situ observation networks (GCOS, 2011). To make it clearer, we added the Bojinski et al. (2014) reference which describes ECVs.

Bojinski, S., Verstraete, M., Peterson, T. C., Richter, C., Simmons, A., Zemp, M.: The concept of Essential Climate Variables in support of climate research, applications, and policy. Bulletin of the American Meteorological Society, 95(9):1431-1443, doi:10.1175/BAMS-D-13-00047.1, 2014.

Change "understorey" by understory :

OK, corrected.

Abstract evaluation: this abstract si unspecific. The abstract should summarize overarching objective, what the project is about and what the main results are. Your abstract does not respond to any of those parameters.

As suggested, the abstract was modified. To follow the reviewer suggestion, we added the following sentence at the end of the abstract:

"This paper aims to present the field work that was carried out during 13 successive rainy seasons, the measured vegetation variables, and the associated open data base. Finally, a few examples of data use are shown.Âż

Line 3: explain what you mean by "one half the total leaf area index"

This definition was adopted by the Global Climate Observing System, GCOS. It corresponds to the one half the total green leaf area per unit horizontal ground surface area (Watson, 1947; Chen and Black, 1992; GCOS, 2011). Leaf should be seen here as a generic term for designing the above-ground areal extent of green vegetation.

. Watson, D.J., 1947. Comparative physiological studies in growth of field crops. I. Variation in net assimilation rate and leaf area between species and varieties, and within and between years. Ann. Bot., 11, 41-76, https://www.jstor.org/stable/42907002. . Chen, J.M and, Black, T.A., 1992. Defining leaf area index for non-flat leaves. Plant Cell Environ., 15(4), 421–429, doi: 10.1111/j.1365-3040.1992.tb00992.x. . GCOS, 2011. Systematic Observation Requirements for Satellite-based Products for Climate, Supplemental Details to the Satellite-based Component of the Implementation Plan for the Global Observing System for Climate in Support of the UNFCCC (update, December).

Line 4-off: References should go in chronologic order.

The references have been modified according to your recommendations.

Line 15: what are production efficiency models?

Production efficiency models are well known models, based on the theory of light use efficiency which states that a relatively constant relationship exists between photosynthetic carbon uptake and radiation absorption at the canopy level. Production efficiency models have been widely used in agronomy and more recently using available satellite data, to estimate Net Primary Production at different spatial scales.

To make it clearer, we added the following reference which describes efficiency models used in combination with satellite observations.

McCallum, I., Wagner, W., Schmullius, C., Shvidenko, A., Obersteiner, M., Fritz, S., et al.: Satellite-based terrestrial production efficiency modeling. Carbon Balance and Management, 4:8, doi: 10.1186/1750-0680-4-8, 2009.

In general acronyms are expressed in parenthesis. Western African Monsoon (WAM)

As suggested, this has been changed.

Line 43: check if you can use an internet link as reference.

Yes, web pages can be given as references under the form: Title, URL.

Normally at the end of an introduction the paper should describe a roadmap of what the paper is about. Missing here.

As suggested, the end of the introduction has been modified as:

ÂńThe main objective of the current field monitoring is to document the seasonal, interannual and decadal variations of LAI and thus primary productivity in relation to the rainfall variability, at different spatio-temporal scales. More precisely, this monitoring aims to provide relevant data to investigate how the WAM and its spatio-temporal variability affect the vegetation density and cover in Central Sahel. This data set can also

be used in the interpretation of Sahelian surface processes that are mainly modulated by the vegetation cover such as the partition between latent and sensible energy fluxes (Timouk et al., 2009), dust and biogenic chemical emissions from soils (Pierre et al., 2012; Delon et al., 2015). In the following, we describe the experimental protocol used for the estimation of vegetation variables, the associated open data base and show a few examples of data use."

Lines 11 and 20: there is a mismatch in the use of supersite and then super-site. You should keep coherency across the paper.

The correction has been made and we changed "supersite" to "super-site".

Lines 25 – off change numbers by the actual word.

As suggested, the requested corrections have been made.

Line 39: change "metre" by meter :

As suggested, the correction has been made.

Line 22: moderate remote sensing sensors. You need to put examples and references here.

A suggested, we modified the sentence by adding the examples of three moderate remote sensing satellites, MODIS, SPOT-VGT and PROBA-V. These satellites have also been indicated in the Introduction section.

Line 25: here you express variations of estimates of LAI and PAI. From where these number so are coming from?

As indicated by the given reference (Mougin et al., 2014), the accuracy of the LAI was estimated in a previous study which detailed the whole methodology for the herbaceous canopies. We use the same approach in the present study for the forest PAI. This is now more clearly specified in the text:

"The accuracy associated to the field LAI estimates is approximately ±17.3% for the herbaceous canopies (Mougin et al., 2014). Following the same approach and by taking into account the errors of classification and spatial sampling, the accuracy on the Acacia forest PAI is approximately ±36.5% (this study).Âż

Line 31: please use a more academic verb instead of "catch"

Following your recommendation, 'catch' has been changed to 'characterize'.

Section 4: what is this? I really don't understand the purpose of this section.

We just follow the guidelines of the journal. The description of the data base and the different measured variables, and also, the availability of the data via an open access, are mandatory for submitting a 'data paper'.

Figure 1. This illustrate very-well the sites but it is difficult to see where this is located. Can you please add a larger area map with an inset so that we can see where this region is actually located.

Following your recommendation, this figure has been modified and a larger map has been added. The new figure 1 is as follows (see the attached file):

Figure 1. The validation sites used in this study: a) Location of the 50 km x 50 km AMMA super-site in Mali, West-Africa, and b) map (on a Google-Earth image, 2016/31/12) of the super-site showing the location of the 8 vegetation canopies: Agoufou (#17, Ag), Timbadior (#18, Ti), Hombori Hondo (#19, Hh), Tara (#31, Ta), Kelma forest (#21, Kf) and Kelma plain (#21b, Kp), Eguerit (#40, Eg), Bilantao (#41, Bi), the national meteorological station at Hombori and the automatic meteorological, flux and soil moisture stations installed during the AMMA project during the 2005-2010 period. The vegetation sites refer to Hiernaux et al. (2009a).

Figure 3. time series of LAI. These series are not clearly explained why Kelma herbs peaks far away from Agoufou?

The 'Kelma herbs' site corresponds to the herbaceous canopy of the seasonally inundated forest site (see Section 2) which contrasts strongly with the other herbaceous sites located on sandy dunes. This explains why the LAI peak is not time coincident with those of sandy sites. These differences are explained in the text (Section 3, P6 L2-4):

"The strong temporal dynamics of the herbaceous cover justifies the 10-day resolution sampling to precisely characterize the seasonal vegetation cycle, particularly the maximum LAI, LAImax, which is usually observed towards the end of the rainy season. Afterwards, on sandy soils the herbaceous green canopy composed of annual plants, rapidly dried out. In the forest site, on clay soils temporarily flooded, the understorey did not dry as fast and is partly composed of a perennial grass Sporobolus helvolus. Âż

Figure 4. Here you show rainfall and LAI. However LAI is known to be very sensitive to temperature and therefore inversely dependent to rainfall. Then for datasets that focus on vegetation that eventually respond to climate shift it it mandatory to know the time series of rainfall rather than the accumulated rainfall. The vegetation onset depends not only on radiation and temperature but also on the precipitation during the beginning of the season.

It is true that LAI is very sensitive to temperature in temperate and boreal regions. Under the dry tropics, air temperature has little effect on vegetation growth which is largely controlled by soil moisture availability in the rooting zone. Accordingly, we agree that time series of rainfall are essential to interpret vegetation growth. However, in a first approximation, the annual cumulated rainfall can be used to interpret inter-annual variations of herbaceous vegetation mass.

Figure 5. Kelma Forest exhibits a different dynamic behavior than Agoufou for LAI below 1 m2m-2." No explanation in the text what is the attribution to this feature. Also the scattering dispersion is larger in the second plot compared to the first plot for LAI

larger than 1 m2m-2.

This figure shows the comparison between field LAI measurements and LAI estimated by the MODIS sensor. We reran the analysis since we found a numerical conversion error when extracting the MODIS data. Figure 5 has been changed by the following figure. The observed scattering dispersions are related to the complexity of the observed canopies (much higher for the open Acacia forest than for the savannah).

Figure 6. It is difficult to understand the meaning of this comparison. I think you should clearly explain why this comparison is needed. And. Actually why the series diverge differently.

Again, this is an illustration on the use of LAI data, here to estimate the performance of land surface models. Using a meteorological forcing and soil and vegetation characteristics, such models can simulate the time variation of vegetation density. The estimation of model performances requires accurate field measurements such as the ones presented in this paper.

Please also note the supplement to this comment:
https://www.earth-syst-sci-data-discuss.net/essd-2018-113/essd-2018-113-AC1-supplement.pdf

[revised manuscript text omitted]

| | 2005 | 2006 | 2007 | 2008 | 2009 | 2010 | 2011 |
|---|---|---|---|---|---|---|---|
| | | | | | 2017/11/19
2017/11/29
2017/12/08
2017/12/17
2017/12/30 | | |
| **Kelma Herbs EW (#21)** | 2005/06/26
2005/07/03
2005/07/22
2005/08/18
2005/08/30
2005/09/18 | 2006/07/24
2006/08/12
2006/08/26
2006/09/12
2006/09/20
2006/10/01
2006/10/17
2006/11/22 | 2007/07/15
2007/07/28
2007/08/03
2007/08/13
2007/08/26
2007/09/08
2007/09/19
2007/09/29
2007/11/17 | 2008/07/08
2008/07/16
2008/07/26
2008/08/10
2008/08/19
2008/08/29
2008/09/05
2008/09/21
2008/09/29 | 2009/08/17
2009/08/27
2009/10/07
2009/11/04 | 2010/07/09
2010/08/09 | 2011/07/22
2011/08/09
2011/08/19 |

| | 2012 | 2013 | 2014 | 2015 | 2016 | 2017 | |
|---|---|---|---|---|---|---|---|
| | 2012/07/25
2012/08/06
2012/08/17
2012/08/26
2012/09/07
2012/09/19
2012/09/30
2012/10/13
2012/10/24
2012/10/30
2012/11/10 | 2013/08/30
2013/09/12
2013/09/21
2013/09/30
2013/10/11 | 2014/08/21
2014/08/28
2014/09/06
2014/09/15
2014/09/26
2014/10/07
2014/10/17 | 2015/07/20
2015/07/30
2015/08/12
2015/08/22
2015/08/31
2015/09/10
2015/09/18
2015/09/28
2015/10/08
2015/10/23
2015/11/02 | 2016/08/17
2016/08/31
2016/09/19
2016/09/29
2016/10/13
2016/10/20
2016/10/27
2016/11/07
2016/11/18
2016/11/28 | 2017/08/04
2017/08/14
2017/08/23
2017/09/06
2017/09/17
2017/09/27
2017/10/07
2017/10/17
2017/10/28 | |

| | 2005 | 2006 | 2007 | 2008 | 2009 | 2010 | 2011 |
|---|---|---|---|---|---|---|---|
| **Kelma Plain EW (#21b)** | - | - | - | 2008/08/10 | 2009/08/07
2009/08/17
2009/08/27
2009/09/16 | - | 2011/07/22
2011/08/09
2011/08/19 |

| | | | | | 2009/10/07 | | |
|---|---|---|---|---|---|---|---|
| | **2012** | **2013** | **2014** | **2015** | **2016** | **2017** | |
| | 2012/06/17 | 2013/06/05 | 2014/08/05 | 2015/07/11 | 2016/07/20 | 2017/06/19 | |
| | 2012/06/30 | 2013/07/07 | 2014/08/21 | 2015/07/20 | 2016/07/29 | 2017/06/28 | |
| | 2012/07/07 | 2013/07/18 | 2014/08/28 | 2015/07/30 | 2016/08/10 | 2017/07/05 | |
| | 2012/07/25 | 2013/08/01 | 2014/09/06 | 2015/08/10 | 2016/08/31 | 2017/07/16 | |
| | 2012/08/06 | 2013/08/10 | 2014/09/15 | 2015/08/21 | 2016/09/19 | 2017/07/24 | |
| | 2012/08/17 | 2013/08/20 | 2014/10/26 | 2015/08/31 | 2016/09/29 | 2017/08/04 | |
| | 2012/08/26 | 2013/08/30 | | 2015/09/10 | | 2017/08/14 | |
| | 2012/09/07 | 2013/09/12 | | 2015/09/18 | | 2017/08/23 | |
| | 2012/09/19 | 2013/09/21 | | 2015/09/28 | | 2017/09/06 | |
| | 2012/09/30 | 2013/09/30 | | | | 2017/09/17 | |
| | 2012/10/13 | 2013/10/11 | | | | 2017/09/27 | |
| **Tara NESW (#31)** | **2005** | **2006** | **2007** | **2008** | **2009** | **2010** | **2011** |
| | - | - | 2007/08/02 | 2008/07/08 | 2009/07/09 | 2010/07/09 | 2011/07/22 |
| | | | 2007/08/16 | 2008/07/16 | 2009/07/17 | 2010/08/09 | 2011/08/09 |
| | | | | 2008/07/23 | 2009/07/29 | | 2011/08/19 |
| | | | | 2008/07/26 | 2009/08/12 | | |
| | | | | 2008/08/11 | 2009/08/15 | | |
| | | | | 2008/08/18 | 2009/08/28 | | |
| | **2012** | **2013** | **2014** | **2015** | **2016** | **2017** | |
| | 2012/09/07 | 2013/06/05 | 2014/08/16 | 2015/07/09 | 2016/07/03 | 2017/06/18 | |
| | 2012/09/18 | 2013/07/06 | 2014/08/28 | 2015/07/20 | 2016/07/20 | 2017/06/28 | |
| | 2012/09/29 | 2013/07/18 | 2014/09/06 | 2015/07/30 | 2016/07/29 | 2017/07/05 | |
| | | 2013/08/01 | 2014/09/15 | 2015/08/10 | 2016/08/10 | 2017/07/16 | |
| | | 2013/08/20 | 2014/09/26 | 2015/08/21 | 2016/08/31 | 2017/07/24 | |
| | | 2013/08/30 | 2014/10/07 | 2015/08/31 | 2016/09/19 | 2017/08/04 | |
| | | 2013/09/12 | 2014/10/17 | 2015/09/10 | 2016/09/29 | 2017/08/14 | |
| | | 2013/09/21 | | 2015/09/18 | | 2017/08/23 | |
| | | 2013/09/30 | | 2015/09/28 | | 2017/09/06 | |
| | | | | | | 2017/09/17 | |
| | | | | | | 2017/09/27 | |

| | | | | | | | 2017/10/07 | |
|---|---|---|---|---|---|---|---|---|

| **Tara NWSE (#31)** | **2005** | **2006** | **2007** | **2008** | **2009** | **2010** | **2011** |
|---|---|---|---|---|---|---|---|
| | - | - | - | 2008/08/18
2008/08/25
2008/09/05 | 2009/08/28 | - | - |
| | **2012** | **2013** | **2014** | **2015** | **2016** | **2017** | |
| | - | - | - | - | - | - | |
| **Eguerit EW (#40)** | **2005** | **2006** | **2007** | **2008** | **2009** | **2010** | **2011** |
| | - | - | - | 2008/08/20 | - | - | - |
| | **2012** | **2013** | **2014** | **2015** | **2016** | **2017** | |
| | **-** | **-** | **-** | **-** | **-** | **-** | **-** |
| **Bilantao NESW (#41)** | **2005** | **2006** | **2007** | **2008** | **2009** | **2010** | **2011** |
| | - | - | 2007/07/30 | 2008/08/17
2008/08/26 | 2009/09/01 | - | - |
| | **2012** | **2013** | **2014** | **2015** | **2016** | **2017** | |
| | - | 2013/07/13 | - | - | - | - | |

---

## Author Comment (AC2) · 12 Apr 2019

Reviewer #2:

First, I would like to appreciate the efforts that have been taken by the authors to collect and analyze 13-years field measurements for the manuscript "A leaf area index, LAI, data set acquired in Sahelian rangelands (Gourma, Mali) over the 2005-2017 period". Secondly, the open-access dataset provided by the authors will be an asset for future research use, which is quite appreciable indeed.

Thank you for your positive appreciation.

[Figure]

However, I believe the manuscript requires major revisions mostly about the way it is written and the structure and text body of especially abstract, introduction, site description, and field sampling strategy and data description. It gets sometimes very difficult to follow the logic and methodology used in this manuscript. I also recommend the authors to fix the sentence structure, e.g., avoid run-on sentences, and grammatical and spelling issues that I have noticed throughout the paper.

Thank you for your valuable comments and suggestions of. We improved the manuscript according to your recommendations.

In addition, I would recommend the authors to rerun the data analysis using yearly MODIS imagery corresponding the year of collected data in the field, which I will explain it later in my list of comments.

Thank you for your advice. In fact, we detected a problem with the data we extracted from the MODIS site, due to a numerical conversion error we made. This problem has been fixed and the corresponding figures (Fig. 5a and 5b) have been modified (see below).

Furthermore, the current type of provided line numbers makes addressing comments very difficult.

Sorry for that but we followed the current type of line numbers provided by the journal.

Here are some of my concerns and comments regarding this manuscript :

1. The title doesn't carry out the main picture and purpose of this research. Also, as I found the study region is in the northern Sahel and thus the title needs to be changed as ". . .rangelands in the northern Sahel. . ."

This paper aims to present briefly the long term measurements of LAI performed in the Gourma region in Mali, and the associated open data base that was built for disseminating the data to the research community. Our motivation was also to show a few examples of data use.

Field measurements were performed in Northern Mali, between Mopti and Gao, and for many authors, this region belongs to the Sahel zone proper (e.g. Le Houérou, 1989) from a biogeographical point of view. Accordingly, we suggest to keep the term 'Sahel' but, as suggested, we modified the title as:

ÂńA leaf area index data set acquired in Sahelian rangelands of Gourma in Mali over the 2005-2017 periodÂż.

Le Houérou H.N., 1989. The grazing land ecosystems of the African Sahel. Ecological Studies 75, Springer-Verlag, Berlin, 295p.

2. Abstract, P 2, L5-10, you can't draw a strong conclusion about using this dataset for a better understanding of the Sahelian vegetation response to the current rainfall changes unless you scientifically provide evidence that this study area at the northern Sahel has similar vegetation community with the rest of the Sahel.

Yes, you agree. We are aware that the Sahel region is not as homogeneous as it is often described. However, due to its climate, soil types and vegetation characteristics similar to many other Sahelian regions, the Gourma site was selected as one of the two Sahelian study sites within the frame of the international African Monsoon Multidisciplinary Analysis project (Lebel et al., 2010). It was also selected in 2010 by the international 'Tropical Biomes in Transition' project, TROBIT, (www.geog.leeds.ac.uk/TROBIT), to be representative of the Sahelian zone of West-Africa. Field studies supported by NASA have also been conducted in the 1980s, to support satellite observations of the whole Sahel (e.g. Hiernaux and Justice, 1986).

We added the following sentence in the Introduction section: "In 2010, the Gourma site was also selected as the Sahelian pilot site of West-Africa for the international 'Tropical Biomes in Transition' project (TROBIT) (www.geog.leeds.ac.uk/TROBIT). Âż

. Lebel et al., 2010. The AMMA field campaigns: Multiscale and multidisciplinary observations in the West African region. QJRMS AMMA Special Issue, 136, 8-33, doi:

[Figure]

doi.org/10.1002/qj.486. . Hiernaux P. and Justice C.O., 1986. Suivi du développement végétal NOAA au cours de l'été 1984 dans le Sahel malien. Int. J. Remote Sensing, 11, 1515,1531.

3. 1. Introduction, L5, please add the exact internet link for this reference (GCOS, 2011) in the citation list and look at the following reference and include it, Bojinski et al. 2014. The Concept of essential climate variables in support of climate research, applications, and policy. Bull. Am. Meteorol. Soc. 95:1431-1443.

Following your suggestion, we added the internet link for GCOS (2011) in the reference list. We also added the Bojinski et al. (2014) reference in the Introduction.

GCOS. Systematic Observation Requirements for Satellite-based Products for Climate, Supplemental Details to the Satellite-based Component of the Implementation Plan for the Global Observing System for Climate in Support of the UNFCCC (update, December), 2011, available at https://library.wmo.int/doc_num.php?explnum_id=3710.

4. P 3 L10, "The seasonal variation of the ECVs, i.e. the vegetation phenology" please change it to "...., i.e. changes in the vegetation phenology".

As suggested, this has been changed (P4, L11).

5. P3 L20, "low trees" is vague. Does it mean short stature trees or dispersed trees?

To make it clearer, this has been changed (P4 L22) to: "scattered stands of shrubs and low trees".

6. P3 L35, MODIS needs to be in parentheses. Please add all abbreviations in the parenthesis and be consistent about using your terminology.

OK, corrected.

7. P3 L35, "...instruments have been providing continuous estimation of the ECV...". Please change it to "... have been used to provide a continuous estimation....".

As suggested, this has been changed.

8. P3 L35-40, You mentioned past research emphasized on using spatially heterogenous rangeland for validating ECVs on for example MODIS imagery, but at the beginning of next page, you are describing your study site as having lowest spatial heterogeneity. Looks these statements are contradictory. Please explain why your selected study site was different from what literature pointed out about the spatial heterogeneity

Field sites characterized by a high spatial heterogeneity are a big challenge for validation studies using moderate resolution (between 500 m to 1 km) instruments, due to the difficulties associated with field sampling at the scale of the satellite resolution, and also because of non-linear effects. The herbaceous vegetation canopies of the Gourma sites are heterogeneous at a fine spatial scale, between a few meters to tens and hundreds of meters, but are homogeneous at a broader scale larger than the satellite resolution which is a major advantage because of the satellite pointing errors.

The sentence was rephrased as: "Among the selected validation sites, the Gourma site stands out as the one with the lowest spatial heterogeneity at the resolution of moderate resolution sensors (Garrigues et al., 2006). This is a major advantage for validation studies because of the ground geolocation uncertainties of satellite products (Garrigues et al., 2008).Âż

9. P3 L40, the whole sentence from "With the main objectives....gov/) projects." is a vague run-on sentence. Please break it down to smaller clear sentences. Also, is it the objective of this study? If so, you need to clearly state this objective. I assume you have two main objectives: 1- validation of computed vegetation indices from the satellite imagery such as MODIS 2-providing open-access dataset. You need to deliver these main objectives at the end of introduction in a clear and strong way.

As suggested, we rephrased the sentence as (P4 L41):

"In 2000, the Sahelian Gourma sites have been integrated among the site network

used in the Validation of Land European Remote Sensing Instruments (VALERI) project (Baret et al., 2005; www.avignon.ina.fr/valeri/; Camacho et al., 2013) and the Committee on Earth Observation Satellites (CEOS)/Land product validation (Morisette et al., 2006; http://lpvs.gsfc.nasa.gov/) projects. Among the selected validation sites, the Gourma site stands out as the one with the lowest spatial heterogeneity at the moderate spatial resolution scale. This is a major advantage for validation studies because of geolocation uncertainties of satellite products (Garrigues et al., 2008)."

10. P4 before L5, "Before the set-up of a seasonal …" this sentence is about field sampling strategy and needs to go to that section.

OK, as suggested, this sentence has been moved to the field sampling section (Section 3).

11. 3. Site description, P4 L10, please change "the annual rainfall mean" to "the mean annual rainfall".

OK, corrected.

12. P4 L20, what is "super-site"? Please explain it at the first time you are using it.

The super-site is a 50 km x 50 km area in which all the validation sites are located. The Hombori super-site is shown in Figure 1 and described in Section 2.

13. P4 L20, "clayed loamy" should be "clay loam".

OK, corrected.

14. P4 L20, "…and its understory herb layer in a clayed-loamy plain, …". Please change it to "...and its understory herb layer in a plain of dominantly clay loam soils,…".

OK, corrected.

15. P4 L25, "clay loamy" and on the next sentence "clay-loamy". Please be consistent about using your terminology and change them to "clay loam".

OK, corrected.

16. P.4, L35, why were you bias about your sampling directions at N-S or E-W, and not to randomly choose the transect direction? Please justify it. 17. P.4, L35,

The N-S or E-W directions were originally chosen for seasonal and long term monitoring of the vegetation canopies (herbs and trees). The corresponding 1 km transects were set perpendicular to the main orientation of the vegetation pattern. For fixed dune systems, it means perpendicular to the dune and interdune succession. Along the 1-km transects, this allows spatial heterogeneity to be taken into account. The choice of the sampling line is described in Hiernaux et al. (2009) and Mougin et al. (2009).

To make it clearer, we modified the text as (P6, L5-9):

Âń These photographs were collected during the growing season of herbaceous canopies or over the whole year for the forest canopy (Figure 2a, 2c), and regularly taken along one or two perpendicular 1 km sampling transects crossing the site, perpendicular to the main orientation of the vegetation pattern that is perpendicular to the dune and interdune succession, usually in the North-South or the East-West directions (Table S3). Along the 1-km transects, this allows spatial heterogeneity to be taken into account. The choice of the sampling lines is described in Hiernaux et al. (2009).

. Hiernaux, P., Mougin, E., Diarra, L., Soumaguel, N., Lavenu, F., Tracol, Y., Diawara, M.: Rangeland response to rainfall and grazing pressure over two decades: herbaceous growth pattern, production and species composition in the Gourma, Mali. J. Hydrol. 375 (1-2), 114–127, doi:10.1016/j.jhydrol.2008.11.005, 2009.

"geolocated" please change it to "georeferenced".

OK, corrected.

18. P.4, L35, "to within an approximately 10-metre accuracy" please change it to "with approximately 10-meter horizontal positional accuracy".

OK, corrected.

19. P4, L40, "approximately 100 (50)"? 100 (50) is vague. Please explain it.

We mean 100 photographs for the 1-km transects and 50 photographs for the 500 m transect in the Kelma forest. To make it clearer, the sentence has been changed to:

ÂńAt each sampling date, 100 or 50 hemispherical photographs were acquired at the 1 km herbaceous or 0.5 km forest sites, respectively, that is a picture taken every 10 meters.Âż

20. P4, L40, "herbaceous (forest) sites" is confusing. Please explain it. The term forest in the parenthesis gives readers the impression of equality to the herbaceous.

This has been changed. See above.

21. P5 L5, "…maintained horizontal thanks to bubble-levels." is vague. What does "thanks" mean here? Please rewrite it in a clear way.

OK, we changed the sentence to Âńmaintained horizontally using a bubble-levelÂż.

22. P5 L10, "non green" please change it to "non-green".

OK, changed.

23. P5 L10, "and has proved to be efficient" please change it to "and has been proved to be efficient".

OK, changed.

24. P5 L25, "±17.3%" and "±36.5%" are accuracy or standard deviations? If they are standard deviations, please be careful about using your terminology.

These are accuracies estimated for the savannah sites in Mougin et al. (2014). For Kelma forest, the accuracy was estimated in the present study using the same approach as for the herbaceous sites. The corresponding sentence was slightly modified as:

Âń The accuracy associated to the field LAI estimates is approximately ±17.3% for the herbaceous canopies (Mougin et al., 2014). Following the same approach and by taking into account the errors of classification and spatial sampling, the accuracy on the Acacia forest PAI is approximately ±36.5% (this study). Âż

25. P5 L30, "Afterwards, the herbaceous green canopy rapidly dried out apart from the forest understory composed partly with the perennial herb . . ..". It is confusing. Please rewrite it in a clear way.

The difference between the vegetation cycles of the sandy sites and the inundated understory comes from the soil moisture conditions and because annual herbs dominate in the sandy sites whereas the forest understory is mainly composed of perennial herbs which exhibit a longer vegetation cycle. The sentence was modified as (P7 L7-9):

ÂńAfterwards, the herbaceous green canopy composed of annual plants on sandy soils, rapidly dried out apart from the seasonally inundated Kelma forest understory, composed partly with the perennial herb Sporobolus helvolus, which benefited from more humid conditions.Âż

26. 5. Use of the data set, P6 L30, this is an objective of this study and needs to go to the end of introduction.

As suggested, this part was moved to the end of the introduction. After the modifications, the section 5 starts by :

ÂńIn addition to the documentation of the seasonal, inter-annual and decadal variations of vegetation variables in relation to rainfall variability, examples of data use in validation exercises of remote sensing products and surface models, are briefly presented below.Âż

27. 4.1. Validation of satellite products, P7, L5, ". . .the clumping efforts performed thanks to the estimated aggregation factor, . . ." what is the meaning of "thanks" here. It is a confusing sentence. Please rewrite it.

We modified the sentence as : "...the clumping effects performed using the estimated aggregation factor, ..."

28. P7 L10, why is "VEGETATION" capitalized? Does it convey a specific meaning here? Please explain it.

Yes, it is because 'VEGETATION' is the name of a moderate resolution satellite dedicated to the monitoring of global vegetation. To avoid any confusion, we replaced 'VEGETATION' by 'SPOT-VEGETATION' as in the Introduction section.

29. P7 L10-15, what does it mean "The collection 6 benefited from improved surface reflectances and biome type inputs (Wolfe et al., 2013), and provided more accurate products (Xu et al., 2018b)"? It is confusing. Why didn't you compare year by year computed vegetation indices from MODIS imagery and field dataset for 13 years? How do you deal with the yearly fluctuations of precipitation in such an arid climate that significantly impacts the leaf area index and other vegetation characteristics? Based on your analysis, you are assuming the vegetation indices derived from MOIDS imagery at the period of 2005-2017 are similar to the period of 2015-2017. I recommend you rerun the analysis using year by year comparison between computed vegetation indices from the MODIS imagery and the field survey.

Sorry for the confusion. We extracted MODIS LAI products that correspond to the exact location of the validation sites and for the date of field observations ($\pm 4$ days). Field and MODIS LAI values were then compared together and a statistical analysis was made. Two examples of such a comparison are shown in Figures 5a and 5b for the savannah and forest sites, respectively. Only the MODIS pixels flagged as 'good' pixels are retained for the comparison. This explains why we finally kept a limited number of MODIS values in the comparison.

30. P6 L20, please change "meso scale" to "mesoscale".

The word 'mesoscale' has been suppressed in this section.

31. P14, Figure 1, Please make a joint map with this figure that shows the study area in the African continent.

OK, as suggested, Figure 1 has been modified to include a small figure showing the location of the validation site in Mali, within the West African region.

32. P18, Figure 5, the issue of specified comparison of MODIS imagery with field surveys, for vegetation indices, is pretty much clear on the data points where data are clumped row by row, especially in the regression plot for Kelma forest, and this type of comparison is not statistically true and can cause erroneous results.

As explained above, we extracted again the MODIS LAI products and select only those issued from the main inversion algorithm. Then, we reran the statistical analysis. The results of the direct comparison study are shown in the corrected Figure 5:

33. P22, Table 3, Why are almost half of your sampling sites include a very low yearly sample size, e.g., sites no. #17 Agoufou NS, #18 Timbadior NS, #18 Timbadior EW, #31 Tara NW-SE, #40 Eguerit EW, and #41 Bilantao NE-SW, and how do you justify drawing conclusion from the results of such an small sample size? What is the reason for not having enough samples at those sites?

The main reason is the occurrence of the Northern Mali conflict (Mali civil war) that started from 2012. A peace deal between the Malian government and rebels was signed on June 2013, but the region is still very unsecured due to the activities of islamists groups, and various armed groups, sometimes promoted by ethnic rivalries. Moreover, drug trafficking is very active. Accordingly, field work remains very risky, and can only be performed during short periods of peace.

However, from 2005, a total of 658 sampling transects have been monitored, and approximately 52 000 hemispherical images have been acquired, processed and analysed. This data set is of considerable interest in a weakly sampled region strongly impacted by the current global change.

By having such a small sample size, making conclusion for those sites with specific plant communities and thus a broader conclusion for the entire vegetation communities across the northern Sahel could be questionable.

This data set complements well numerous past and recent studies in the Gourma region (see for instance the papers: Hiernaux et al., 1984; Hiernaux et al., 2009a; 2009b; Mougin et al., 2009 ; Dardel et al., 2014). Associated to vegetation observations collected in other Sahelian countries (for instance in Senegal by the Institute of Geography of Copenhagen, in Niger by the French National Research Institute for Development), the present data set can be very useful to support regional studies at the Sahel scale.

. Hiernaux P., Cisse, M. I., & Diarra, L., 1984. Bilan d'une saison des pluies 1984 tres deficitaire dans le Gourma (Sahel malien): Première campagne de suivi et télédétection expérimentale. Addis Ababa, Ethiopia: ILCA. . Hiernaux P., Lassine D., Trichon V., Mougin E., Baup F., 2009c. Woody plant population dynamics in response to climate changes from 1984 to 2006 in Sahel (Gourma, Mali). Journal of Hydrology, AMMA CATCH special issue, 375(1-2), 103-113, doi: 10.1016/j.jhydrol.2009.01.043. . Hiernaux P., Mougin E., Diarra L., Soumaguel N., Lavenu F., Tracol Y., Diawara M., 2009a. Rangeland response to rainfall and grazing pressure over two decades: herbaceous growth pattern, production and species composition in the Gourma, Mali. Journal of Hydrology, AMMA CATCH special issue, 375(1-2), 114-127, doi: 10.1016/j.jhydrol.2008.11.005. . Mougin E., Hiernaux P., Kergoat L., Grippa M., et al., 2009. The AMMA Gourma observatory site in Mali: Relating climatic variations to changes in vegetation, surface hydrology, fluxes and natural resources. Journal of Hydrology, AMMA CATCH special issue, 375(1-2), 14-33, doi:10.1016/j.jhydrol.2009.06.045. . Dardel C., Kergoat L., Hiernaux P., Mougin E., Grippa M., Tucker C.J., 2014. Re-greening Sahel: 30 years of remote sensing data and field observations (Mali, Niger). Remote Sensing of Environment, 140, 350-364, doi :10.1016/j.rse.2013.09.011.

34. Supplements, Figure S2, the picture numbering (the letters) doesn't correspond to

the letters written in the figure caption.

Thank you for noticing. This has been corrected.

35. The type of soils in Table S1 and Table S2 do not look correspond to each other for some sites. For example, Kelma Plain (#21b) site shows a dominant soil type of clay loam in the Table S1 but the percentage of soil particles in Table S2 shows a very low amount of clay particles about 1-2 %. Please check them out and use the common soil textural triangle to find out the soil texture class for every site

Thank you again for noticing. For the Kelma plain, the type 'clay loam' has been changed to 'silt loam' in accordance with the soil textural triangle. The Tables S1 and S2 look correct for the other sites.

**b)**

15.50°N — 🟩 Eg

**a)** MALI

500 km

15.33°N — Hh  Ag  Ti  Bi  Homboridi

**Legend**

🟩 vegetation site

⊻ raingauge

✚ AMMA automatic weather station

✚ AMMA flux station

🔻 AMMA soil moisture station

⚫ Town of Hombori

Ta  Kp  Kf

15.17°N

5 km

1.67°W          1.50°W          1.33°W

**Fig. 1.** Figure 1. The validation sites used in this study: a) Location of the 50 km x 50 km AMMA super-site in Mali, West-Africa, and b) map (on a Google-Earth image, 2016/31/12) of the super-site showing the

[Figure]

[Figure]

**Fig. 2.** Figure 5. Comparison between field LAI or PAI estimates and MODIS Collection 6 LAI products for the period 2005-2017: a) Agoufou herbaceous canopy and b) Kelma forest site. In this later case, the MOD

---

## Author Response (AR2)

**Topical Editor Decision: Publish subject to minor revisions (review by editor)** (21 Apr 2019) by Falk Huettmann
Comments to the Author:
Fairbanks Alaska, 18th April 2019

Dear Editors, Colleagues,

Thanks for the final version and submission with a detailed reply! I am happy to provide my review for ESSD-2018-113. Overall, the authors did a nice and great job to reply and address concerns expressed by the two reviewers. It's GOOD. However, a few typos and small errors are found in the rebuttal letter, as well as in the MS text. Considering the importance of this MS, its data and coverage, this looks somewhat careless and so I kindly ask to correct the ones in the MS text (see below for a list). Again, they are minor but they will make the MS and data much better.

Once done, we can move ahead into production and approve the MS for publication. I congratulate the authors and thank them for their work. This is a relevant MS and data set indeed.

Yours Falk Huettmann, Associate Editor

University of Alaska Fairbanks (UAF)

Thank you for your positive appreciation.

List of a few typos and English corrections in the MS text:

Thank you for the suggestions of corrections. In the new revised version of the manuscript, the modified sentences have been highlighted in yellow.

Abstract: LAI needs to be mentioned as a term first and then followed by the abbreviation. Please correct.

It is what we did in the first submitted version of the manuscript. Fot the revised version, we followed the suggestion of the first reviewer who gave the following recommendation:

*«Through the **abstract** there are three instances in which you define acronyms that are not used through the abstract. - then I would suggest to use the entire word and then when the paper start in the Introduction you define those acronyms.»*

Accordingly, we removed all the abbreviations from the abstract and then defined the acronyms in the Introduction section. Were we wrong ?

First phrase of the MS is very clumsy; please correct and use proper English; ideally two phrases.

As suggested, the first phrase was modified as:

*«The Global Climate Observing System, GCOS, identified the leaf area index, LAI, as one of the main terrestrial essential climate variables, ECVs, to be monitored from systematic long-term satellite and in situ measurements (GCOS, 2011; Bojinski et al., 2014). The LAI corresponds to the one half the total green leaf area per unit horizontal ground surface area (Watson, 1947; Chen and Black, 1992). Similarly, the plant area index, PAI, corresponds here to the above-ground areal extent of green*

*vegetation. Both LAI or PAI are related to the fraction of absorbed photosynthetic active radiation (0.4–0.7 μm) by the green vegetation, fAPAR (Myneni and Williams, 1994; Fensholt et al., 2006), and to the canopy green cover fraction, fCover, i.e. the amount of green vegetation distributed in a horizontal plane (Carlson and Ripley, 1997).»*

Line 9: needs plural of ECV

- corrected

Line 17: 'on' reads odd in that phrase, please correct.

The phrase

*«The Sahel is a vast ecoclimatic and biogeographic region extending south of the Sahara* on *6000 km long and 400-600 km wide, from the Atlantic Ocean to the Red Sea (Le Houérou, 1989) »*

has been changed to:

*«The Sahel is a vast ecoclimatic and biogeographic region extending south of the Sahara* over *6000 km long and 400-600 km wide, from the Atlantic Ocean to the Red Sea (Le Houérou, 1989)»*

Line 43: in situ should likely be in italics.

- Corrected

Page 5 line 15 and line 30: paragraph spacing looks odd, please check and make consistent as needed.

- Checked and corrected

Throughout the MS text authors refer to LAI and always use both: the full term and the abbreviation. I suggest to introduce LAI once and then use the abbreviation only. It's consistency what is needed here. Please check and correct; can likely be done easily with find-replace.

- Checked and corrected

Page 7: Section 4.1 needs to be 5.1; please check and correct.

- Thank you for noticing, corrected.

Page 7: Section 4.2 needs to be 5.2; please check and correct.

- Thank you again for noticing, corrected.

Conclusion: Reads like an abstract but not a conclusion. Please focus on the conclusion (what is concluded from this work ?!). I suggest to change a phrase or two to achieve it.

As suggested, we changed the last sentence of the conclusion:

*«Among others, this data set can be used to validate moderate resolution satellite products, to establish relationships between satellite vegetation indices and vegetation variables, to evaluate land surface models, to interpret surface fluxes, and above all, in association with other data sets to study the decadal response of semi-arid ecosystems to the current climatic change.*

Acknowledgements: This is in the discretion of the authors, but I would always thank the reviewers (with name or without), it's more transparent and considerate that way. Anyways, that's up to the authors.

As suggested, the acknowledgements have been changed to :

*«The two reviewers and the topical Editor who provided helpful suggestions and comments on earlier versions of this manuscript are gratefully acknowledged.»*

Figure 1: Legend needs to be consistent (Town should be in lower case then)

On the map of Figure 1, the name 'Hombori' has been changed to 'HO'. We hope it corresponds to your recommendation.

Figure 5: Both have exactly the same R2, yes ?

The actual R2 values are approximately the same, more precisely R2=0.7367 for Agoufou, and R2=0.7403 for Kelma. Accordingly, we decided to give the same value '0.74' on both figures.

Finally, in my view, generalizing from a few sites to all of Mali etc should be done and stated very carefully – specifically when not modeled. So perhaps authors can add a phrase to that effect; just to make sure.

The objective will be more to extend these preliminary results to the Sahel belt and to similar semi-arid ecosystems than to the whole Mali which exhibits from North to South, very different climate and ecosystems, with the Sahara desert to the North, to the tropical moist forests in the South.

We slightly modified the end of the conclusion by adding more general perspectives related to the monitoring of the 'greening' Sahel.

*«Among others, this data set can be used to validate moderate resolution satellite products, to establish relationships between satellite vegetation indices and vegetation variables, to evaluate land surface models and to interpret surface fluxes. More specifically, this data set can be useful to interpret the current greening trend that is observed widely over the Sahel since the 1980s' drought.»*

Non-public comments to the Author:
NA, but well done!